# Strain Transfer Function of Distributed Optical Fiber Sensors and Back-Calculation of the Base Strain Field

**DOI:** 10.3390/s21103365

**Published:** 2021-05-12

**Authors:** Sangyoung Yoon, Meadeum Yu, Eunho Kim, Jaesang Yu

**Affiliations:** 1Division of Mechanical System Engineering, Jeonbuk National University, Jeonbuk 54896, Korea; sangyoung563@gmail.com (S.Y.); 201950278@jbnu.ac.kr (M.Y.); 2Automotive Hi-Technology Research Center & LANL-CBNU Engineering Institute-Korea, Jeonbuk National University, Jeonbuk 54896, Korea; 3Multifunctional Structural Composite Research Center, Institute of Advanced Composite Materials, Korea Institute of Science and Technology (KIST), Jeonbuk 55324, Korea; jamesyu@kist.re.kr

**Keywords:** distributed optical fiber sensor, strain transfer function, strain back-calculation, structural health monitoring

## Abstract

Distributed optical fiber sensors are a promising technology for monitoring the structural health of large-scale structures. The fiber sensors are usually coated with nonfragile materials to protect the sensor and are bonded onto the structure using adhesive materials. However, local deformation of the relatively soft coating and adhesive layers hinders strain transfer from the base structure to the optical fiber sensor, which reduces and distorts its strain distribution. In this study, we analytically derive a strain transfer function in terms of strain periods, which enables us to understand how the strain reduces and is distorted in the optical fiber depending on the variation of the strain field. We also propose a method for back-calculating the base structure’s strain field using the reduced and distorted strain distribution in the optical fiber sensor. We numerically demonstrate the back-calculation of the base strain using a composite beam model with an open hole and an attached distributed optical fiber sensor. The new strain transfer function and the proposed back-calculation method can enhance the strain field estimation accuracy in using a distributed optical fiber sensor. This enables us to use a highly durable distributed optical fiber sensor with thick protective layers in precision measurement.

## 1. Introduction

The demands for structural health monitoring of large-scale structures keep increasing, and various technologies are developing, such as distributed sensors covering wide areas [1,2,3,4], sensor networks with the help of wireless communications [5], etc. Distributed optical fiber sensors (DOFS) are one of the promising technologies for monitoring the structural health of large-scale structures. They show high potential not only for structural health monitoring, but also for sensing temperature, acoustic waves, electric and magnetic fields, or chemicals, amongst other applications, owing to attractive features such as lightweight, small size, immunity to electromagnetic interference, and non-corrosive features [2,6]. Recently, optical fiber sensors attract great attention in measuring strain and temperature changes in cryogenic environments [7], particularly for various applications using superconductors [8,9,10,11,12,13,14,15,16,17,18], cryogenic pressure vessels [19], etc. Moreover, DOFS and quasi-distributed optical fiber sensors (q-DOFS) expand their application field to wearable devices [20], integrated sensor-actuators [21], and biomedical areas [22,23]. Their applications for structural health monitoring include oil and gas pipelines [24], civil structures [25], rail tracks [26], aircraft structures at the sub-coupon level [27], and composite cylinders [28,29]. Various sensing mechanisms using distributed optical fibers have been developed, and their performance continues to improve [2].

Most distributed optical fiber sensors made of silica materials are fragile and are easy to break during handling, installation, and service. Thus, packaging or coating with nonfragile materials is commonly used to protect them [30,31,32,33,34,35]. However, the packaging and coating can reduce and distort the strain field transferred from the base structure to the optical fiber. Decreasing the thickness of the protecting layer and adhesive layer is desirable for reducing the signal distortion, which, however, reduces the sensor’s durability. Thus, improving both the durability and accuracy of the distributed optical fiber sensors can be challenging. Researchers have investigated the effects of packaging, coating, and bonding layers on the strain transfer for more accurately interpreting the strain data measured from the optical fiber sensor and designing the packaging and coating layers [32,33,34,35,36]. Most of these studies have been performed using local fiber optic sensors (e.g., the Fiber Bragg Grating (FBG) sensor). A short FBG sensor bonded (or embedded) onto (into) the base structure measures an average strain of the local area. The sensing component of the FBG sensor (i.e., the grating location) should be located far enough from the boundary effect region where strain significantly varies. Other than the boundary effect region, the strain transmission ratio remains constant under a constant strain field in the base structure. Various strain transmission ratios have been theoretically derived based on simplified models with various intermediate layers, boundary conditions, installation types (i.e., surface bonding or embedding) [35,36,37,38,39,40]. This has been well summarized in a recent review paper [41]. A few studies have considered more complex geometry for more realistic analysis, such as side width, top and bottom thickness of the adhesive layer [42], and many-layered structure (7 layers) with extended boundary conditions [43].

In contrast to the local sensors, distributed optical fiber sensors can sense varying strain fields in a wide area, for instance, varying strain fields around the impact damage of a composite structure [28]. In the varying strain field, the strain transfer may not occur at a constant ratio. A few studies have investigated strain transfer from a varying strain field to the distributed optical fiber sensors. Feng et al. [44] have theoretically investigated and experimentally validated the strain transfer mechanism of a distributed optical fiber sensor bonded onto a base structure with a crack. Henault et al. [45] have proposed a mechanical transfer function for a distributed optical fiber embedded in a concrete structure to relate the measured and actual strains near a crack in concrete structures. Tan et al. [46] have studied strain transfer of a distributed optical fiber sensor embedded in a structure and considered an arbitrary strain field using polynomial functions. Duck and LeBlanc [47] have considered a spatial wavelength-dependent strain transfer for a simple embedded optical fiber sensor model and have shown that the transfer coefficient depends on wavelength.

In this study, we adopt the spatial wavelength-dependent strain transfer approach [48] to consider an arbitrary strain field in the strain transfer for a surface bonded distributed optical fiber. We derive an equation for fiber strain from a simplified model consisting of an optical fiber, coating, adhesive, and a base structure with a harmonically varying strain field. From the equation, we obtain a strain transfer function (STF) which expresses how the base structure’s strain transfers to the optical fiber depending on strain periods. We also newly propose a method for back-calculating the base strain from the optical fiber strain using the strain transfer function. The analytical strain transfer function and the accuracy of the back-calculation method are verified with strain transfer simulations using a finite element method.

The remainder of the paper is structured as follows. First, in Section 2.1 we derive the strain transfer function based on a simplified model, and we explain the back-calculation method in Section 2.2. We then describe the numerical analysis models in Section 3. In Section 4, we validate the strain transfer function and explain the effects of various parameters on strain transfer. We also demonstrate the back-calculation of the base strain from the distorted fiber strain using a composite beam model with an open hole. In Section 5, we propose a way to enhance both the durability and accuracy of distributed optical fiber sensors and conclude the paper.

## 2. Analytical Model for Strain Transfer Analysis

### 2.1. Strain Transfer Function

In this section, we analytically derive the strain transfer function for a distributed optical fiber sensor. Our analysis uses a simplified model consisting of an optical fiber sensor (OFS), a coating layer, an adhesive layer, and a base structure, as shown in Figure 1, and is based on several assumptions: (i) all materials are linear elastic, (ii) all layers bond perfectly with neighboring layers, (iii) the OFS shows only axial deformation, (iv) the adhesive layer shows only shear deformation, and (v) an external load is applied only to the substrate. This model has been used to derive a surface-bonded optical fiber sensor’s strain transmission ratio under constant strain distribution [34,35,36]. Assumption (i) and (ii) are reasonable when the optical fiber sensor is well fabricated/installed without defects between layers, and it deforms in a small strain range in the elastic regime. Assumption (iii) and (iv) apply when the OFS diameter is relatively small and the adhesive layer is relatively soft compare to the other parts (coating, OFS, and base structure). Assumption (v) becomes reasonable when the dominant deformation of the base structure is tension/compression without (local) bending deformation.

Here we use the same type of model to obtain an analytical expression of strain transfer function under a periodic strain distribution. This function is then used for accurately back-calculating the strain distribution on the surface of a base structure.

For the strain transfer function, we begin by deriving a differential equation for fiber strain ($\varepsilon_{f}$) from a series of equilibrium relations. From force equilibrium in the coating layer in Figure 1, we obtain the following equation:(1)$$r_{c}\int_{0}^{\pi }T_{ca}(\theta ,z)\;d\theta -r_{f}\int_{0}^{2\pi }T_{fc}(\theta ,z)\;d\theta +\pi (r_{c}^{2}-r_{f}^{2})\frac{d\sigma_{c}(r,\theta ,z)}{dz}=0$$
where $r_{f}$ and $r_{c}$ are radiuses of fiber and coating layer, respectively. $T_{ca}(\theta ,z)$ and $T_{fc}(\theta ,z)$ are shear stresses at the interface between the coating layer and the adhesive layer, and at the interface between the fiber and coating layer, respectively. In other words, the first and second terms represent the shear forces at the coating-adhesive and the coating-fiber interfaces, respectively. The third term represents the axial force of the coating layer and $\sigma_{c}(r,\theta ,z)$ is the axial stress of the coating layer. The shear force at the fiber-coating interface $T_{fc}(\theta ,z)$ (the second term in Equation (1)) can be expressed in terms of the axial stress of fiber ($\sigma_{f}$) from force equilibrium in fiber.
(2)$$-\int_{0}^{2\pi }T_{fc}(\theta ,z)\;d\theta =\frac{d\sigma_{f}}{dz}\pi r_{f}$$
we assume here that the fiber deforms only in the axial direction, without shear deformations. This is reasonable when relatively stiff fiber with a small diameter is attached to the base using relatively soft adhesive materials (see Table 1).

Next, we relate $T_{ca}(\theta ,z)$ to the base displacement ($u_{b}$). The shear stress in the coating layer ($\tau_{c}$) at an arbitrary radius r is expressed as below from an equilibrium relation,
(3)$$\tau_{c}(r,\theta ,z)=\frac{r_{c}}{2r}T_{ca}(\theta ,z)+\frac{r_{c}^{2}-r^{2}}{2r}\frac{d\sigma_{c}}{dz}\;(r_{f}<r<r_{c})$$

From Hooke’s law ($\tau_{c}=G_{c}\gamma_{c}$) and the strain-displacement relation ($\gamma_{c}=\partial u_{c}/\partial r=\tau_{c}/G_{c}$), the relative displacement at a position r in the coating layer($u_{c}$) is obtained by integrating the shear strain ($\gamma_{c}$) from $r_{f}$ to r. The displacement at the radius r in the coating layer becomes Equation (4) by adding the interface displacement between fiber and coating $U_{fc}(\theta ,z)$.
(4)$$u_{c}(r,\theta ,z)=U_{fc}(\theta ,z)+\frac{1}{G_{c}}\left[\frac{r_{c}}{2}\ln\left(\frac{r}{r_{f}}\right)T_{ca}(\theta ,z)+\frac{1}{2}\left(r_{c}^{2}\ln\left(\frac{r}{r_{f}}\right)-\frac{1}{2}(r^{2}-r_{f}^{2})\right)\frac{d\sigma_{c}(r,\theta ,z)}{dz}\right]$$

We next define shear stress in the adhesive layer ($\tau_{a}$):(5)$$\tau_{a}(\theta ,z)=T_{ca}(\theta ,z)=G_{a}\frac{u_{b}(z)-U_{ca}(\theta ,z)}{t(\theta )}$$
where $u_{b}$ is the displacement at the base and t(θ) is the thickness of the adhesive layer in the vertical direction, expressed as $t(\theta )=t_{\max}-r_{c}\sin\theta$ (see Figure 1). The displacement at the coating-adhesive interface is calculated from Equation (4) $U_{ca}(\theta ,z)=u_{c}(r_{c},\theta ,z)$, which is introduced to Equation (5) to obtain the shear stress at the coating-adhesive interface:(6)$$T_{ca}(\theta ,z)=f(\theta )\left[\left(u_{h}-U_{fc}\right)-\frac{1}{2G_{c}}\left(r_{c}^{2}\ln\left(\frac{r_{c}}{r_{f}}\right)-\frac{1}{2}(r_{c}{}^{2}-r_{f}^{2})\right)\frac{d\sigma_{c}(r,\theta ,z)}{dz}\right]$$
where $f(\theta )=G_{a}G_{c}/[G_{c}(t_{\max}-r_{c}\sin\theta )+\{G_{a}r_{c}\ln(r_{c}/r_{f})\}/2]$ is defined in the range 0≤θ≤π, since only the bottom half of the coating layer is attached to the adhesive layer (see Figure 1).

We introduce Equations (2) and (6) into Equation (1) and differentiate Equation (1) in terms of z, we obtain the equation
(7)$$\begin{array}{l} r_{c}\int_{0}^{\pi }f(\theta )\left[\left(\frac{du_{h}}{dz}-\frac{dU_{fc}}{dz}\right)-\frac{1}{2G_{c}}\left(r_{c}^{2}\ln\left(\frac{r_{c}}{r_{f}}\right)-\frac{1}{2}(r_{c}{}^{2}-r_{f}^{2})\right)\frac{d^{2}\sigma_{c}(r,\theta ,z)}{dz^{2}}\right]\;d\theta \\ \;+\pi r_{f}{}^{2}\frac{d^{2}\sigma_{f}}{dz^{2}}+\pi (r_{c}^{2}-r_{f}^{2})\frac{d^{2}\sigma_{c}(r,\theta ,z)}{dz^{2}}=0 \end{array}$$

The modulus of fiber ($E_{f}$) and coating ($E_{c}$) are at similar levels and are far greater than the adhesive modulus (see Table 1). Thus, we can assume that the variation in the axial strain of fiber and coating layer is almost the same $d\varepsilon_{f}/dz≅d\varepsilon_{c}/dz$. Moreover, if we substitute $du_{b}/dz=\varepsilon_{b}$, $dU_{fc}/dz=\varepsilon_{f}$, $\sigma_{f}=E_{f}\varepsilon_{f}$ and $\sigma_{c}=E_{c}\varepsilon_{c}$ in Equation (7), we obtain the equation
(8)$$\begin{array}{l} E_{f}\frac{d^{2}\varepsilon_{f}}{dz^{2}}\left[\pi r_{f}^{2}-r_{c}\int_{0}^{\pi }f(\theta )\frac{E_{c}}{2E_{f}G_{c}}\left(r_{c}^{2}\ln\frac{r_{c}}{r_{f}}-\frac{r_{c}^{2}-r_{f}^{2}}{2}\right)d\theta +\pi \left(r_{c}^{2}-r_{f}^{2}\right)\frac{E_{c}}{E_{f}}\right]\\ \;+r_{c}\int_{0}^{\pi }f(\theta )d\theta \;\left(\varepsilon_{b}(z)-\varepsilon_{f}(z)\right)=0 \end{array}$$

This is written simply as a 2nd order ordinary differential equation in terms of fiber strain, as
(9)$${\varepsilon^{\prime \prime }}_{f}(z)-\lambda^{2}\varepsilon_{f}(z)=-\lambda^{2}\varepsilon_{b}(z)$$
where $\lambda^{2}$ is a constant defined as
(10)$$\lambda^{2}=\frac{r_{c}\int_{0}^{\pi }f(\theta )d\theta \;}{E_{f}\pi r_{f}^{2}-\frac{r_{c}E_{c}}{2G_{c}}\left(r_{c}^{2}\ln\frac{r_{c}}{r_{f}}-\frac{r_{c}^{2}-r_{f}^{2}}{2}\right)\int_{0}^{\pi }f(\theta )d\theta \;+\pi \left(r_{c}^{2}-r_{f}^{2}\right)E_{c}}$$

If we assume that the base strain ($\varepsilon_{b}$) varies harmonically along z-direction with a period of 2π/a, $\varepsilon_{h}(z)=C\cos(az)$, then a general solution of Equation (9) becomes
(11)$$\varepsilon_{f}(z)=A\cosh\lambda z+B\sinh\lambda z+C\frac{\lambda^{2}}{\lambda^{2}+a^{2}}\cos az$$

The first two terms depend on the boundary conditions at both edges of the fiber, while the third term represents the strain transferred from the base structure. Therefore, the strain transfer function (STF), which represents the strain transfer from the base structure to the fiber, becomes Equation (11) in the region where the boundary effect is negligible.
(12)$$STF(a)=\frac{\varepsilon_{f}}{\varepsilon_{b}}=\frac{\lambda^{2}}{\lambda^{2}+a^{2}}$$

This demonstrates that the strain transfer function depends on the spatial period (2π/a) of the base strain. As the period of the base strain decreases, the strain transmission correspondingly decreases. This strain transfer function is useful for understanding how an arbitrary base strain is transferred to the optical fiber depending on the base strain’s spatial period. We numerically validate the strain transfer function (Equation (11)) in Section 4 using a finite element method.

### 2.2. Back-Calculation Method of The Base Strain Field

In this section, we explain the back-calculation of the base strain from the strain in the distributed optical fiber sensor.

An arbitrary 1D strain field signal can be expressed as a combination of multiple strain periods. We can get the strain periods using the Fourier transform of the signal. When an arbitrary strain field containing multiple strain periods is transmitted to the optical fiber, the strain field may distort due to the different strain transmission ratio for each strain period (see Equation (11)) included in the strain field. Here we propose a method for back-calculating the base strain field using the strain transfer function in Equation (11). Its procedure is shown in Figure 2. First, we take a Fourier transform of the fiber strain ($\varepsilon_{OFS}(z)$) using a fast Fourier transform (FFT) algorithm. The transformed data are expressed in a wavenumber (inverse of the spatial period) domain. We then multiply the inverse of the strain transfer function to the fiber strain data ($\varepsilon_{OFS}(\omega )$) in the wavenumber domain, which recovers the reduced strain amplitude for each strain period. Finally, we perform an inverse Fourier transform of the data using an inverse fast Fourier transform (IFFT) algorithm to obtain the base strain ($\varepsilon_{Base}(z)$) in the spatial domain. Notably, the upper wavenumber limit of the FFT data is 1/2Δz, here Δz is the spatial interval of the data. Thus if we decrease the data interval (Δz) in the space domain, we can consider more precise strain variation with small periods in the back-calculation. Additionally, increasing the number of data with a given spatial interval makes the FFT data denser in the wavenumber domain. It makes the strain back-calculation more accurate. Thus, if the data have noise or the data in the spatial domain is not dense enough, one can carefully consider filtering or interpolating the data before using the FFT algorithm. This back-calculation of the base strain is demonstrated with a finite element model in Section 4.4.

## 3. Finite Element Analysis

### 3.1. Validation of Strain Transfer Function

We use the finite element method to validate the strain transfer function derived from the analytical approach (Equation (11)) and to investigate the effects of plastic deformation, material properties, and the thickness of a coating and an adhesive layer on strain transfer from the base to the optical fiber. The same type of FE model has been used in the strain transfer analysis of a metal-coated optical fiber sensor under a constant strain field, and it is validated with experimental data in [36].

The inset of Figure 3 shows the FE model consisting of a fiber, a coating layer, an adhesive layer, and a base structure. The curves in Figure 3 represent the stress-strain relation of the coating material (aluminum (AA1235-O)) and the adhesive material (Epoxy) based on experiment data [36,49], where the yield stress of the coating layer and adhesive layer are both assumed to be 30 MPa. Details of the material properties and dimensions of the model appear in Table 1. The ABAQUS/standard program is used for the FE analysis. The 3D stress elements are used for the FE model; brick elements (C3D8R) for the fiber, coating layer, and base structure, and tetrahedron elements (C3D10) for the adhesive layer.

To validate the strain transfer function, we suppress the torsion and bending deformation of the 3D FE model using boundary conditions. The nodes at the centerline of the base structure along z-direction are suppressed from moving x- and y- direction, while the nodes at the centerline of the fiber along z-direction are suppressed from moving x-direction (see inset of Figure 3). We also use two kinds of boundary conditions (BC) for the surfaces at both ends of the structure. One consists of free and z-symmetric BCs to investigate the strain transfer near the free boundary. The other consists of periodic and periodic BCs along z-direction to investigate the effect of strain period on strain transfer. For the periodic strain distribution along z-direction in the base structure, we directly controlled the displacements of the base structures. We get the displacement field by integrating the target strain distribution as
(13)$$u(\tilde{z})=\int_{0}^{\tilde{z}}D\cos\lambda z\;dz$$
where *D* sets for the maximum strain becoming 0.356%. This corresponds to the maximum stress of 600 MPa in the base structure.

### 3.2. Strain Transfer around the Circular Hole of Composite Plate

We also analyze the strain field around an open circular hole in a composite plate under tension (see Figure 4) and demonstrate its strain transfer to the optical fiber for various coating and adhesive thicknesses. The dimensions of the plate are 100 mm (length) × 25 mm (width) × 1 mm (thickness). It consists of two layers [45°/−45°] with a layer thickness of 0.5 mm in the thickness direction. The material properties of the composite (carbon fiber reinforced plastic, CFRP) are shown in Table 2. The open circular hole with a 5 mm diameter is located at the center of the plate. The optical fiber bonds onto the plate at a distance of 5 mm from the center of the hole (see Figure 4). The material properties of the fiber, coating and adhesive are the same as in Table 1. To apply a tensile load to the plate in a lengthwise direction, we fix one side of the plate and apply tensile displacement to the other side. The mesh in the FE model is small enough to capture the strain variation; particularly in the OFS model, the interval between nodes in length-direction (z-direction) is 0.2 mm, which can consider strain periods larger than 0.4 mm in the strain back-calculation.

Three models with different adhesive and coating thicknesses (10 μm and 18.5 μm;100 μm and 62.5 μm; 200 μm and 137.5 μm) are analyzed, and their strain transmission to the fiber are compared in Section 4.3.

## 4. Results and Discussion

### 4.1. Strain Transfer in Harmonic Strain Distribution

We analyze the strain transfer in a 70 mm length structure with free and z-symmetric boundary conditions, as shown in Figure 5a. We apply a harmonic strain distribution with a 20 mm period to the base structure in a lengthwise direction (red dotted line in Figure 5b), with a maximum strain in both compression and tension of 0.356 %. To investigate the effect of plastic deformation in the coating layer, we compare two analysis models; one considers only elastic deformation (Figure 5b), while the other considers elastoplastic deformations (Figure 5c). The graph at the top left in Figure 5b compares the base strain (red dotted line) and the optical fiber strain (black line). The strain distributions of the base and the optical fiber are almost the same except near the free boundary conditions. The stress of the optical fiber at the free boundary should be zero and do not follow the base strain. Thus, the strain transmission ratio (i.e., the ratio of fiber strain to base strain) near the free boundary converges to zero (see graph at bottom left in Figure 5b). Interestingly, other than near the free boundary, the strain transmission ratio becomes constant when the strain field varies by a single period, thus satisfying the strain transfer function in Equation (11). Figure 5c represents the results taking account of elastoplastic deformations in the coating layer. It should be noted that the adhesive layer does not show plastic deformation within the given strain range (0.356%) (see solid blue lines in Figure 3). The difference in strain transfer compared to the elastic model (Figure 5b) mainly appears around zero strain, as shown in the magnified view in Figure 5c, where the fiber strain amplitude decreases significantly. This is also confirmed by the strain transmission ratio in Figure 5c bottom left. This is because of the highly-strained regions in the coating layer (shaded area in Figure 5c) showing plastic deformation. Suppose part of the coating layer locally shows plastic deformation (shaded area). In that case, its stiffness decreases compared to elastic stiffness (compare the slope of the solid red line (elastic region) with the dotted red line (plastic region) in Figure 3). This makes the coated fiber in the plastic region relatively soft (i.e., the equivalent stiffness decreases) and induces more strain locally in the shaded section. In contrast, the relatively stiff elastic region (unshaded area) became less strained due to greater elongation in the plastic region. Thus, the strain transmission ratio in the plastic region is slightly larger than that of the elastic model, whereas, in the elastic region, it became smaller than that of the elastic model.

### 4.2. Effect of the Spatial Period on Strain Transfer

To investigate the effect of the spatial period of the base strain on strain transfer to the fiber, we analyze various strain fields over different periods from 2 mm to 20 mm. For efficient FE analysis, we use a single-period model with periodic boundary conditions at both ends of the structure. Five strain distributions (Figure 6a) are applied to the base structure, whose corresponding strain transmission ratios in the elastic model are shown in Figure 6b. We observe a constant strain transmission ratio along the fiber length direction, which increases and converges to 1 as the spatial period of the strain increases. In the elastoplastic model (Figure 6c), the overall strain transmission ratio also increases as the strain period increases. However, due to local plastic deformation in the coating layer, the strain transmission ratio varies with the fiber, as explained in the preceding section.

To further investigate the effect of local plastic deformation of the coating layer on strain transfer, we compare three models with different coating thicknesses (62.5, 87.5, 137.5 μm) as shown in Figure 7a. The applied strain amplitude is the same as in the previous analysis (0.356 %), while the spatial period of the strain is 20 mm. As the coating thickness increases, the difference in the strain transmission ratio between the elastic and plastic regions increases, as shown in Figure 7c. This is related to a change in the equivalent stiffness of the coated fiber (a combination of fiber and coating stiffness, see Figure 7b). As the coating thickness increases, the corresponding difference between the equivalent elastic and plastic stiffness becomes larger, which increases strain in the plastic region while decreasing it in the elastic region. This produces a wide discrepancy in the strain transmission ratio, as shown in Figure 7c, and distorts strain distribution in the optical fiber.

### 4.3. Effect of Coating and Adhesive Thickness on Strain Transfer

We also investigate the effect of coating and adhesive thickness on the strain transmission ratio. Figure 8a shows strain transmission ratio in terms of strain period calculated from Equation (11) (lines) and FE analysis (symbols) for the three models with different coating thicknesses (81, 125, 200 μm) and a fixed adhesive layer thickness of 10 μm. The strain transmission ratio decreases as the strain period decrease, as explained in Figure 6. The drop in the strain transmission ratio became more significant as the strain period became smaller. When the coating thickness increases, the overall strain transmission ratio decreases. Adhesive thickness also has similar effects on the strain transmission ratio, as shown in Figure 8b, which shows the strain transmission ratio at various adhesive thicknesses (10, 50, 200 μm) with a fixed coating thickness of 125 μm. This is because local shear deformation in the coating and adhesive layers generally reduces strain transfer to the fiber. Thus, the strain transmission ratio overall decreases as the coating and adhesive thickness increase. The analytical results are slightly larger than those from the FE analysis because of the assumptions used in the derivation process for Equation (11); for example, Poisson’s ratio and axial stress of the adhesive layer are not considered. The FE results considering only elastic deformation correspond closely with the analytical predictions using Equation (11). However, if the applied strain becomes larger than the elastic limit of the coating layer, we need to consider elastoplastic deformation since local variation in the strain transmission ratio may distort strain distribution. In Figure 8c, the square symbol represents the maximum strain transmission ratio in the plastic region, while the circle symbol represents the minimum strain transmission ratio in the elastic region.

Figure 8d compares the strain distribution of the base structure (dashed blue line) with those of the fibers considering only elastic deformation (dashed green line) and elastoplastic deformation (solid red line). Optical fiber strain without considering plastic deformation (dashed green line) produces a shape similar to that for base strain, except that its amplitude decreases due to a small strain transmission ratio of around 0.6. However, optical fiber strain considering elastoplastic deformation (solid red) shows distortion in strain distribution compared to the base strain.

### 4.4. Back-Calculation of the Base Strain in Composite Plate

In this section, we analyze strain distribution around an open hole on the composite plate and its transfers to the optical fiber bonded onto the surface (see Figure 4). We also demonstrate the back-calculation of the base strain from the distorted strain distribution in the fiber using the approach shown in Figure 2. Here we consider only an elastic deformation with small elongation of the specimen, 0.1 mm, which is 0.1% of plate length.

Figure 9a–c compare strain distributions of three models with different coating and adhesive thicknesses to investigate the effect of those thicknesses on strain transfer. When the coating and adhesive thicknesses are small (18.5 μm and 10 μm, respectively), as shown in Figure 9a, the strain distributions both of the composite plate and the optical fiber are almost the same. This is because the strain transmission ratio is close to 1 in this case. However, as the coating and adhesive thickness increases (see Figure 9b,c), the strain distribution in the optical fiber distorts compared to that of the composite plate. Significantly, the distortion became severe near to the open hole, where the strain substantially varies in the spatial domain. This is because the strain distribution near the hole contains multiple short period components, and their strain transmission ratio substantially decreases as the strain period decreases (see Figure 8a,b). This non-uniform strain transmission depending on the strain period (i.e., wavenumber) induces distortion in the strain distribution.

We can back-calculate the base strain distribution by restoring the reduced strain amplitude of each strain period in the distorted strain distribution (OFS strain) following the procedure in Figure 2. Figure 10 compares the strain distribution of the optical fiber (thin solid blue line), the strain distribution from the back-calculation (dotted red line), and the strain of the composite plate (thick solid black line) for the thick coating and adhesive model (Figure 8c). The strain distribution from the back-calculation became very close to that on the surface of the composite plate. We presumed that the minor discrepancy near the peaks between the back-calculated strain and the composite strain in Figure 10 is due to the assumptions used in Equation (11).

It is noted that here we use a simplified model for the validation of the spatial period dependent strain transfer function and the strain back-calculation method proposed. However, for practical applications of the proposed methods, one can use other OFS models [32,33,34,35,36,37,38,39,40,43] depending on the test model to derive the strain transfer function. Additionally, we can consider using empirical approaches [42] together for a better fit to the test model.

## 5. Conclusions

In this study, we analytically derive a strain transfer function in terms of a spatial period of the strain field for a distributed optical fiber sensor bonded onto a base structure. We numerically validate the analytical strain transfer function using the finite element method. The effects of various parameters on strain transfer are investigated, including the effects of local elastoplastic deformation, coating, and adhesive layer thickness, and the spatial period of the strain field. We find that local plastic deformation can induce effective stiffness variation and distort strain distribution in the optical fiber. We also find that strain transfer decreases as the strain period decreases, which is significant in a small period regime. This variation of the strain transmission depending on the strain period can substantially distort a strain field. Here we propose a way to back-calculating the base strain from the distorted strain distribution and numerically demonstrate that it properly restores the base strain.

This back-calculation method with the strain transfer function enables us to enhance both the durability and accuracy of distributed optical fiber sensors. A distributed sensor with a thin protecting layer can have high accuracy but has low durability, while a sensor with a thick protecting layer can have high durability but has low accuracy. We can increase the accuracy of the durable sensor with a thick protecting layer using the proposed back-calculation method. This is useful for designing and determining installation parameters of the distributed optical fiber sensors for precision sensing under harsh environments such as structures exposed to severe vibration, external impacts, high temperature, or cryogenic temperature.

Although this study focuses on the analytical and numerical investigation of the strain transfer function and strain back-calculation method, further studies can include investigation of the temperature effect on the strain transfer and experimental validation of the proposed method.

## Figures and Tables

**Figure 1 sensors-21-03365-f001:**
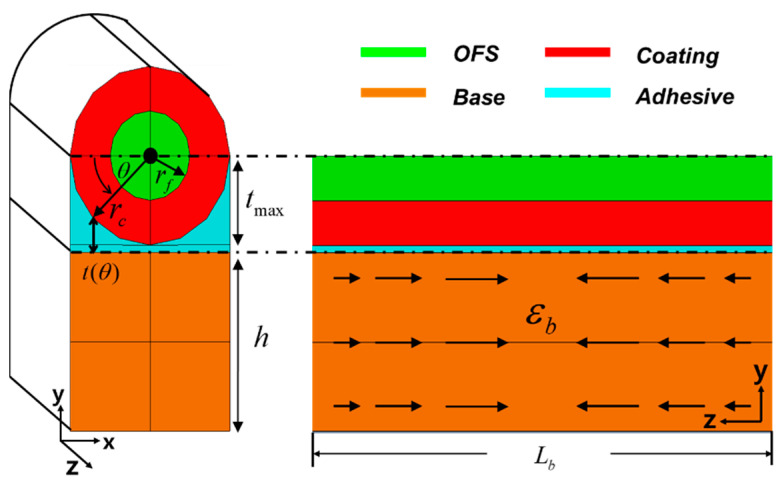
A simplified model for the distributed optical fiber sensor bonded onto a base structure, (**left**) the cross-section in the x-y plane, (**right**) the cross-section in the y-z plane.

**Figure 2 sensors-21-03365-f002:**
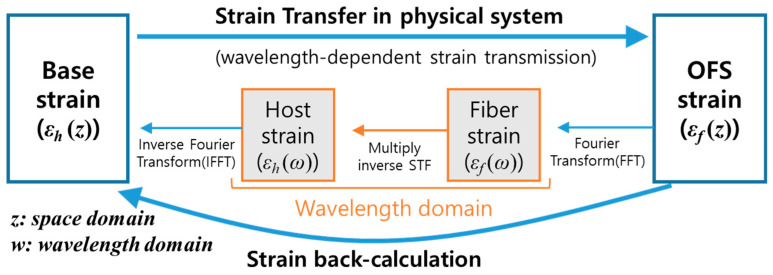
Diagram representing the strain transmission from the base to the optical fiber and the back-calculation of the base strain using the strain transfer function (STF).

**Figure 3 sensors-21-03365-f003:**
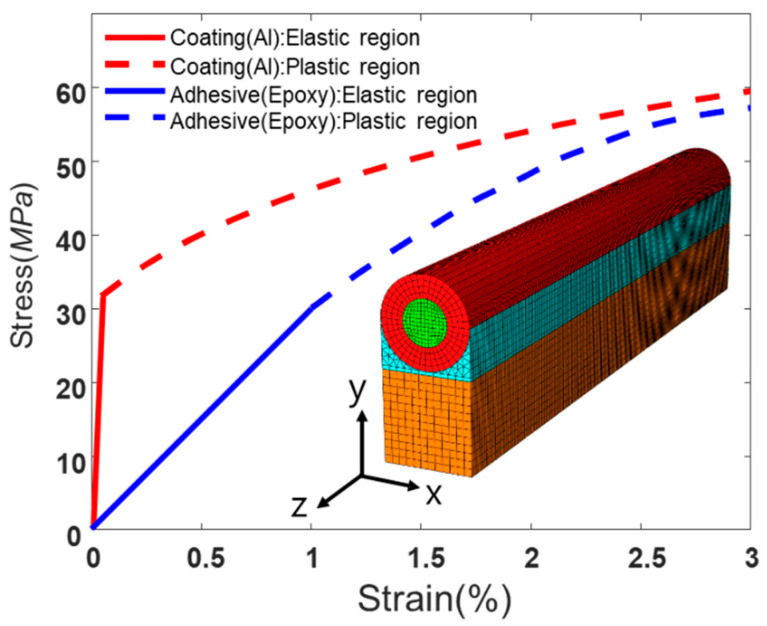
Stress-strain curves of the coating (red line) and the adhesive materials (blue line), solid lines represent linear elastic behavior, and dashed lines represent plastic behavior. The inset shows the FE model of the distributed optical fiber sensor bonded on to the base structure.

**Figure 4 sensors-21-03365-f004:**
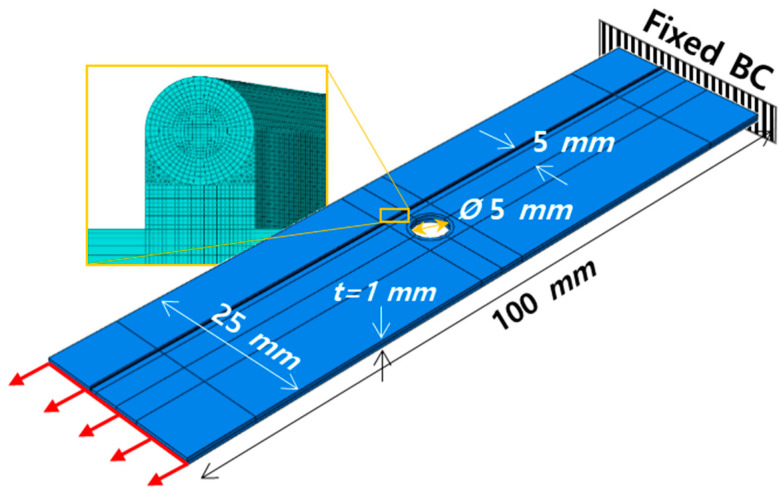
FE model of the composite plate with a circular hole and the magnified view of the distributed optical fiber sensor bonded onto the composite plate.

**Figure 5 sensors-21-03365-f005:**
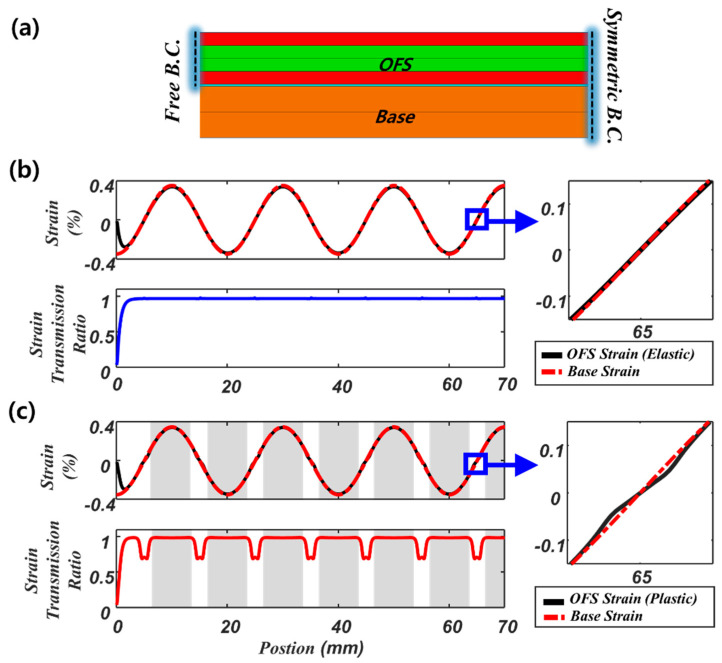
The strain transmission ratio of the optical fiber sensor (70 mm length) with free and symmetric boundary conditions (**a**) cross−section of the FE model, (**b**) strain distributions in the base (red dotted line) and the optical fiber sensor (solid black line), and the corresponding strain transmission ratio (solid blue line) considering only elastic deformation, (**c**) strain distributions in the base (red dotted line) and the optical fiber sensor (solid black line), and the corresponding strain transmission ratio (solid red line) considering elastoplastic deformation.

**Figure 6 sensors-21-03365-f006:**
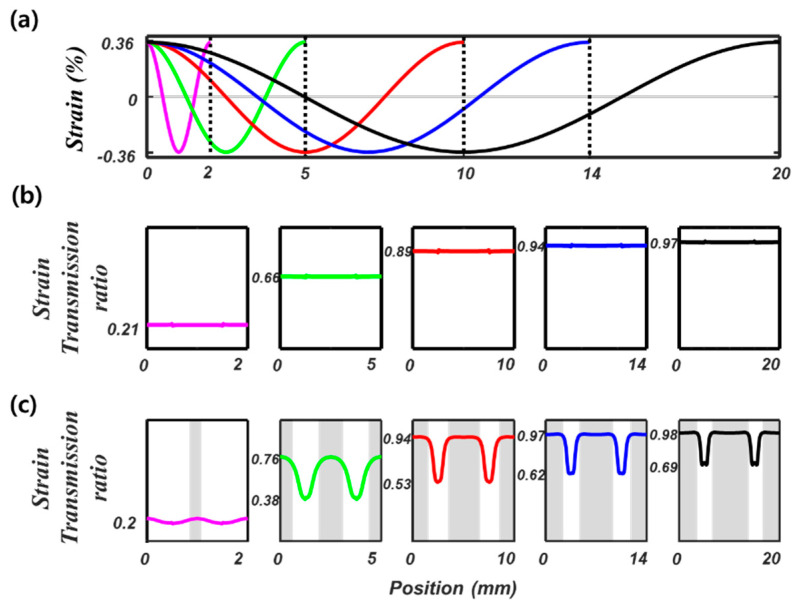
(**a**) Five strain distributions with different spatial periods (2 mm, 5 mm, 10 mm, 14 mm, 20 mm) applied to the base structure (**b**) strain transmission ratios considering elastic deformation (**c**) strain transmission ratios considering elastoplastic behavior of the coating layer.

**Figure 7 sensors-21-03365-f007:**
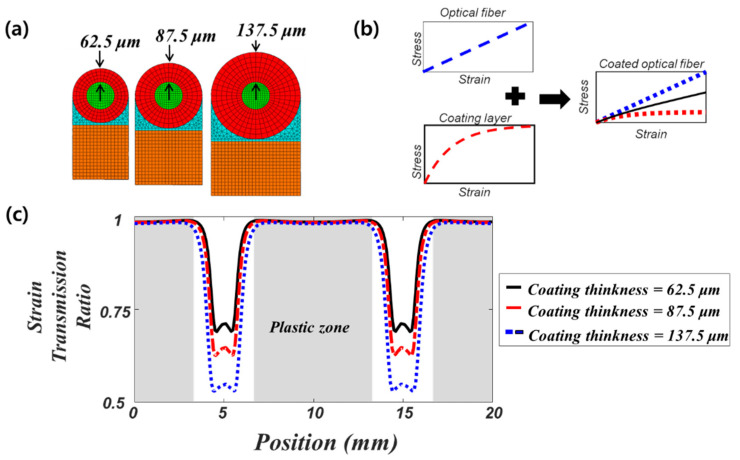
Effect of coating thickness on the strain transmission ratio (**a**) cross-sections of the three models with different coating thicknesses (62.5, 87.5, 137.5 μm) (**b**) illustration explaining the effective stiffness of the coated fiber (**c**) comparison of strain transmission ratios for the three models with different coating thicknesses.

**Figure 8 sensors-21-03365-f008:**
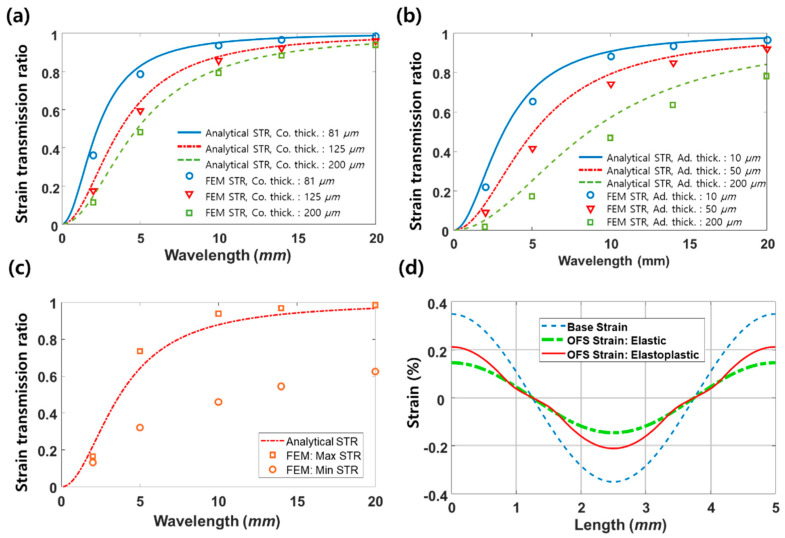
(**a**) Comparison of strain transmission ratios under elastic deformation among three models with different coating thickness (81, 125, and 200 μm); here the adhesive thickness is 10 μm, lines represent analytical prediction, and symbols represent FEA result (**b**) comparison of strain transmission ratios among three models with different adhesive thicknesses (10, 50, and 200 μm) under elastic deformation; here coating thickness is 125 μm, (**c**) the maximum (square) and the minimum (circle) strain transmission ratios taking into account the elastoplastic deformation of the coating layer in FE analysis when coating and adhesive layer thickness are 125 μm and 50 μm, and (**d**) the corresponding strain distributions in the base structure (dashed blue line) and in the OFS under elastic deformation (dash-dot green line) and elastoplastic deformation (solid red line).

**Figure 9 sensors-21-03365-f009:**
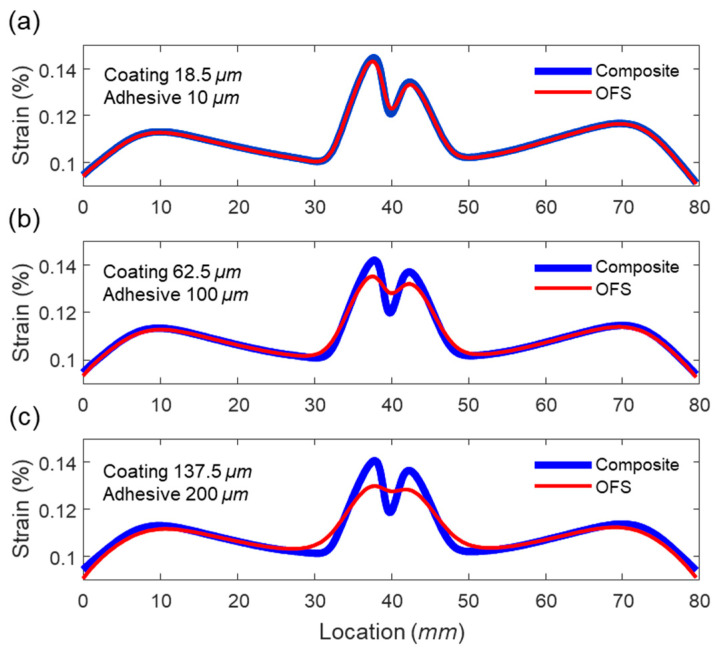
Comparison of composite strain (thick blue line) and OFS strain (thin red line) distributions when (**a**) coating and adhesive thicknesses are 18.5 μm and 10 μm, respectively, (**b**) coating and adhesive thicknesses are 62.5 μm, and 100 μm, respectively, (**c**) coating and adhesive thicknesses are 137.5 μm and 200 μm, respectively.

**Figure 10 sensors-21-03365-f010:**
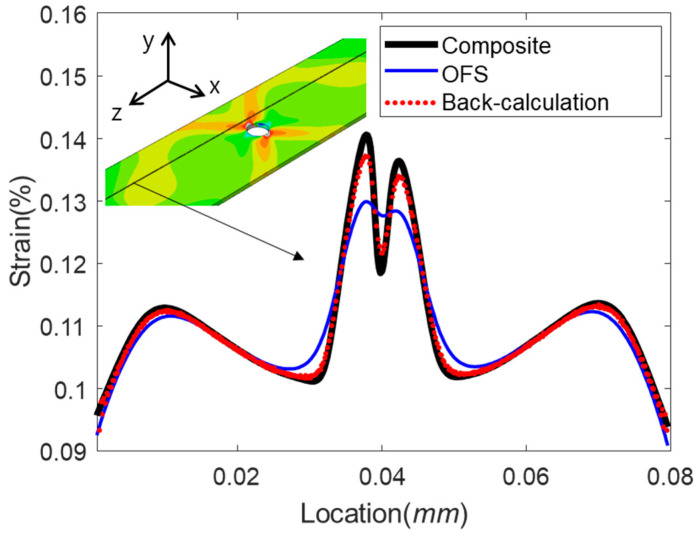
Comparison of strain distributions on the base structure (thick black line), in optical fiber sensor (thin blue line), and the base strain from back-calculation (dotted red line). Inset shows strain distributions on the composite surface along the z-direction.

**Table 1 sensors-21-03365-t001:** Dimensions and mechanical properties of the theoretical model [36,37].

Descriptions	Symbols	Values (Units)
Young’s modulus of OFS	$E_{f}$	73 (GPa)
Poisson’s ratio of OFS	$v_{f}$	0.17
Young’s modulus of coating	$E_{c}$	69 (GPa)
Poisson’s ratio of coating	$v_{c}$	0.33
Young’s modulus of adhesive	$E_{a}$	3 (GPa)
Poisson’s ratio of adhesive	$v_{a}$	0.35
Young’s modulus of base structure	$E_{b}$	168.1 (GPa)
Poisson’s ratio of base structure	$v_{b}$	0.26
Outer radius of OFS	$r_{f}$	62.5 (μm)
Outer radius of coating	$r_{c}$	125 (μm)
Angle in x-y plane	θ	(radian)
Thickness of adhesive layer at θ	t(θ)	(Parameter)
Minimum thickness of adhesive layer (*θ* = π /2)	$t_{\min}$	10 (μm)
Maximum thickness of adhesive layer (*θ* = 0, π)	$t_{\max}$	$t_{\min}+r_{c}$ (μm)
Length of model	$L_{b}$	(Parameter)
Height of base structure	h	0.25 (m)
Base strain	$\varepsilon_{b}$	-

**Table 2 sensors-21-03365-t002:** Mechanical properties of the CFRP plate [49].

Mechanical Properties of the CFRP					
E11	E22, E33	G12, G13	G23	ν12, ν13	ν23
134.6 GPa	7.65 GPa	3.68 GPa	3.2 GPa	0.298	0.52

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
