# Peer review of "Strain Transfer Function of Distributed Optical Fiber Sensors and Back-Calculation of the Base Strain Field"

_sensors, 2021, doi:10.3390/s21103365_

Round 1
Reviewer 1 Report
The following issue should be improved.
In topic 2.2 you explain the approach of Strain back-calculation and mention the fast Fourier transform (FFT) as the used algorithm.
The accuracy of calculations by the using FFT algorithm depends on many factors.
- The value of the sampling (discrete quanta) in the spatial domain.
- The value of the limiting interval in spatial domain (how many signal periods contained by the array in the spatial domain).
- Matching the sampling of the sh(ω) and sampling the sj(ω) in frequency domain .
Please explain the calculation of these values in your simulation.
Author Response
Please see the attachement.

Reviewer 2 Report
Overall, the study and findings of this work are interesting, but please read the following comments that are geared mostly toward improving the readability of the paper. One of the aspects that I would like to be clarified by the authors is about the validation of their numerical models. It appears to me that the FEM model is used to validate the results of the analytical model, but I am not aware of a prior validation of their FEM model (this is mentioned in bullet point 6.).
- In general, the paper needs to be proofread for grammatical errors and awkward language so that the paper is easier to read and has minimal errors.
- Particularly, the sentences on lines 49-52 are not grammatically correct and it is unclear what the sentences are trying to convey. Other sentences within the document should also be reviewed for grammatical mistakes.
- Table 1 should be expanded to also include all the variables shown in Figure 1.
- Additionally, the angle, ?, had contradictory interpretations in Figure 1 compared to Table 1. For example, in the table, tmin is calculated when the angle is measured from the conventionally negative vertical axis, while the figure shows the angle being calculated from the conventionally negative horizontal axis. This should be reviewed such that both the figure and table matches, and the angle is properly interpreted for the equations used.
- Line 96 states that the model used is a "1D model", however, based on the figures and equations, it is unclear how to the model is 1D. Please clarify which model is referred to as 1D, or revise the statement so that it accurately reflects the model used.
- Lines 98-102 list assumptions that I used for this analysis. Please clarify or provide justification as to why these assumptions are appropriate.
- Whether the FEM model has been validated is unclear to me; please include results from the literature, or show in what way you confirmed that the FEM model is accurate. If references were used for validating the FEM, please highlight them more clearly for the reader to understand.
- Please also review the figure captions for grammar and syntax errors.
- Here’s a list of references I deem relevant and should be added, both in the Introduction and in your Results and Discussion section;
- References to add in the Results and Discussion section:
- Wan, Kai Tai, Christopher KY Leung, and Noah G. Olson. "Investigation of the strain transfer for surface-attached optical fiber strain sensors." Smart materials and structures 17, no. 3 (2008): 035037.
- Falcetelli, Francesco, Leonardo Rossi, Raffaella Di Sante, and Gabriele Bolognini. "Strain transfer in surface-bonded optical fiber sensors." Sensors 20, no. 11 (2020): 3100.
- Bastianini, Filippo, Paweł Bocheński, Raffaella Di Sante, Francesco Falcetelli, Leonardo Rossi, and Gabriele Bolognini. "Strain Transfer Estimation for Complex Surface-Bonded Optical Fibers in Distributed Sensing Applications." In 2020 Italian Conference on Optics and Photonics (ICOP), pp. 1-4. IEEE, 2020.
- Di Sante, Raffaella, Lorenzo Donati, Enrico Troiani, and Paolo Proli. "Reliability and accuracy of embedded fiber Bragg grating sensors for strain monitoring in advanced composite structures." Metals and Materials International 20, no. 3 (2014): 537-543.
- Henault, J. M., J. Salin, G. Moreau, M. Quiertant, F. Taillade, K. Benzarti, and S. Delepine-Lesoille. "Analysis of the strain transfer mechanism between a truly distributed optical fiber sensor and the surrounding medium." Concrete Repair, Rehabilitation and Retrofitting III (2012): 733-739.
- Tan, Xiao, Yi Bao, Qinghua Zhang, Hani Nassif, and Genda Chen. "Strain transfer effect in distributed fiber optic sensors under an arbitrary field." Automation in Construction 124 (2021): 103597.
- Please consider adding the following references to the Introduction section to mention cryogenic applications that are emerging as an important field of application of optical fiber sensors (both point sensors such as FBG and distributed sensors)
- Zhou, Kao, Li Ren, Jing Shi, Ying Xu, Dongshen Pu, Guilun Chen, and Yuejing Tang. "Feasibility study of optical fiber sensor applied on HTS conductors." Physica C: Superconductivity and its Applications 575 (2020): 1353693.
- Hu, Qiang, Xingzhe Wang, Mingzhi Guan, and Beimin Wu. "Strain responses of superconducting magnets based on embedded polymer-FBG and cryogenic resistance strain gauge measurements." IEEE Transactions on Applied Superconductivity 29, no. 1 (2018): 1-7.
- Scurti, F., S. Ishmael, G. Flanagan, and Justin Schwartz. "Quench detection for high temperature superconductor magnets: a novel technique based on Rayleigh-backscattering interrogated optical fibers." Superconductor Science and Technology 29, no. 3 (2016): 03LT01.
- Chiuchiolo, Antonella, Luca Palmieri, Marco Consales, Michele Giordano, Anna Borriello, Hugues Bajas, Andrea Galtarossa, Marta Bajko, and Andrea Cusano. "Cryogenic-temperature profiling of high-power superconducting lines using local and distributed optical-fiber sensors." Optics letters 40, no. 19 (2015): 4424-4427.
- Scurti, Federico, and Justin Schwartz. "Optical fiber distributed sensing for high temperature superconductor magnets." In 2017 25th Optical Fiber Sensors Conference (OFS), pp. 1-4. IEEE, 2017.
- Jiang, Junjie, Zeming Wu, Bin Liu, Jie Sheng, Longbiao Wang, Zhuyong Li, Zhijian Jin, and Zhiyong Hong. "Thermal stability study of a solder-impregnated no-insulation HTS coil via a Raman-based distributed optical fiber sensor system." IEEE Transactions on Applied Superconductivity 29, no. 2 (2019): 1-4.
- Scurti, F., J. D. Weiss, D. C. Van Der Laan, and J. Schwartz. "SMART conductor on round core (CORC®) wire via integrated optical fibers." Superconductor Science and Technology 34, no. 3 (2021): 035026.
- Scurti, Federico, Srivatsan Sathyamurthy, Martin Rupich, and Justin Schwartz. "Self-monitoring ‘SMART’(RE) Ba2Cu3O7− x conductor via integrated optical fibers." Superconductor Science and Technology 30, no. 11 (2017): 114002.
- Mizutani, Tadahito, Nobuo Takeda, and Hajime Takeya. "On-board strain measurement of a cryogenic composite tank mounted on a reusable rocket using FBG sensors." Structural Health Monitoring 5, no. 3 (2006): 205-214.
- Lupi, Carla, Ferdinando Felli, Andrea Brotzu, Michele Arturo Caponero, and Antonio Paolozzi. "Improving FBG sensor sensitivity at cryogenic temperature by metal coating." IEEE Sensors Journal 8, no. 7 (2008): 1299-1304.
- Scurti, F., John McGarrahan, and Justin Schwartz. "Effects of metallic coatings on the thermal sensitivity of optical fiber sensors at cryogenic temperatures." Optical Materials Express 7, no. 6 (2017): 1754-1766.
- R. Rajinikumar, K. G. Narayankhedkar, G. Krieg, M. Suber, A. Nyilas and K. P. Weiss, "Fiber Bragg Gratings for Sensing Temperature and Stress in Superconducting Coils," in IEEE Transactions on Applied Superconductivity, vol. 16, no. 2, pp. 1737-1740, June 2006, doi: 10.1109/TASC.2005.864332.
- Van Der Laan, D. C., J. D. Weiss, F. Scurti, and J. Schwartz. "CORC® wires containing integrated optical fibers for temperature and strain monitoring and voltage wires for reliable quench detection." Superconductor Science and Technology 33, no. 8 (2020): 085010.
- Lastly, a suggestion I would like to make to the authors for potential future work is to study how the strain transfer changes at cryogenic temperatures; this would be an interesting topic for the community. For example, comparing the results with the room temperature behavior, it would be interesting to know whether the strain measured by the optical fiber is closer to the actual strain in the host structure at cryogenic temperature than what it is at or around room temperature.
Reviewer 3 Report
This paper numerically demonstrates the back-calculation of the base strain using a composite beam model with an open hole and an attached distributed optical fiber sensor. The new strain transfer function and the back calculation method can enhance the strain field estimation accuracy using a distributed optical fiber sensor. This enables us to use a highly durable distributed optical fiber sensor with protective layers in precision measurement. I have some comments to highlight this paper.
1) Introduction needs improvements in order to highlight recent paper using quasi-distributed optical fiber sensors (q-DOFS) and DOFS) in different fields like wearables, actuators, medical, etc. Please read and refer: a) Quasi-distributed torque and displacement sensing on a series elastic actuator’s spring using FBG arrays inscribed in CYTOP fibers, IEEE Sensors Journal 19 (11), 4054-4061, 2019. b) Wearable and Fully-Portable Smart Garment for Mechanical Perturbation Detection with Nanoparticles Optical Fibers, IEEE Sensors Journal, 21, 2021. c) Spatially resolved thermometry during laser ablation in tissues: Distributed and quasi-distributed fiber optic-based sensing, Optical Fiber Technology, 58, 2020.
2) How this proposed study can be applied on polymer optical fiber? What are the influence and points to take into account? Because in table 1 the authors consider some parameter values from silica fiber like Young modulus and for polymers we have much lower value.
3) The authors claim: A distributed sensor with a thin protecting layer can have high accuracy but has low durability, while a sensor with a thick protecting layer can have high durability but has low accuracy. How we can deal with this study to have the best option? or it is dependent of application? Please add some words.
4) How we can have an idea about this numerical study for a future application as real? I mean, what are the issues we can have if we go for experimental field/fabrication? Please comment.
5) In figure 2 where the authors show a diagram, how we can understand the influence of temperature variation since in wavelength domain we need to take it in account.
Round 2
Reviewer 3 Report
The paper was improved as suggested.